

# The association between prior physical fitness and depression in young adults during the COVID-19 pandemic—a cross-sectional, retrospective study

Yaoshan Dun[1,2,3,*], Jeffrey W. Ripley-Gonzalez[1,*], Nanjiang Zhou[1], Qiuxia Li[1], Meijuan Chen[1], Zihang Hu[1], Wenliang Zhang[1], Randal J. Thomas[3], Thomas P. Olson[3], Jie Liu[4], Yuchen Dong[5] and Suixin Liu[1]

[1] Division of Cardiac Rehabilitation, Department of Physical Medicine & Rehabilitation, Xiangya Hospital of Central South University, Changsha, Hunan, China
[2] National Clinical Research Center for Geriatric Disorders, Xiangya Hospital of Central South University, Changsha, Hunan, China
[3] Division of Preventive Cardiology, Department of Cardiovascular Medicine, Mayo Clinic, Rochester, MN, USA
[4] Hunan Traditional Chinese Medical College, Zhuzhou, Hunan, China
[5] Medical College of Jinhua Polytechnic, Jinhua, Zhejiang, China
* These authors contributed equally to this work.

Corresponding author
Suixin Liu, liusuixin@csu.edu.cn

## ABSTRACT

**Background:** The COVID-19 pandemic has led to a spike in deleterious mental health. This dual-center retrospective cross-sectional study assessed the prevalence of depression in young adults during this pandemic and explored its association with various physical fitness measures.

**Methods:** This study enrolled 12,889 (80% female) young adults (mean age 20 ± 1) who performed a National Student Physical Fitness battery from December 1st, 2019, to January 20th, 2020, and completed a questionnaire including Beck's Depression Inventory in May 2020. Independent associations between prior physical fitness and depression during the pandemic were assessed using multivariable linear and binary logistic regressions accordingly, covariates including age, dwelling location, economic level, smoking, alcohol, living status, weight change, and exercise volume during the pandemic. Sex- and baseline stress-stratified analyses were performed.

**Results:** Of the study population 13.9% of men and 15.0% of women sampled qualified for a diagnosis of depression. After multivariable adjustment, anaerobic (mean change 95% CI −3.3 [−4.8 to 1.8]) aerobic (−1.5 [−2.64 to −0.5]), explosive (−1.64 [−2.7 to −0.6]) and muscular (−1.7 [−3.0 to −0.5]) fitness were independently and inversely associated with depression for the overall population. These remained consistent after sex- and baseline stress-stratification. In binary logistic regression, the combined participants with moderate, high or excellent fitness also showed a much lower risk compared to those least fit in anaerobic (odd ratio (OR) 95% CI 0.68 [0.55–0.82]), aerobic (0.80 [0.68–0.91]), explosive (0.72 [0.61–0.82]), and muscular (0.66 [0.57–0.75]) fitness.

**Conclusions:** These findings suggest that prior physical fitness may be inversely associated with depression in young adults during a pandemic.

# INTRODUCTION

Since its onset, coronavirus-2019 (COVID-19) has personified overwhelming stresses, stemming from—infections, loss of work, loss of freedoms, isolation, and death, leaving behind an aura of uncertainty amongst the world's citizens. COVID-19 has led to a spike in deleterious mental health issues (*Wang et al., 2020*), particularly in depression (*Brooks et al., 2020*). There's concern that such a surge may lead to an increased rate of suicides (*Reger, Stanley & Joiner, 2020*) domestic abuse, economic, and somatic health issues (*Moulton, Pickup & Ismail, 2015*). Hence, evidence-based research is needed to address the rise in depression across multiple populations as a result of COVID-19.

Physical fitness, particularly cardiorespiratory fitness, and exercise training have repeatedly been shown to be negatively associated with future CVD and diabetes risk (*Dun et al., 2019a*, *2019b*; *Myers et al., 2015*; *Pedersen et al., 2019*) as well as having a positive relationship with mental health in multiple populations (*Baumeister et al., 2017*; *Cho et al., 2019*; *Kerling et al., 2015*). Studies have been published on COVID-19 and previous outbreaks and their adverse effects on mental health (*Brooks et al., 2020*; *Torales et al., 2020*). Furthermore, recent studies have alluded to an association between current physical activity and mental health during the pandemic (*Lopez-Bueno et al., 2020*; *Stanton et al., 2020*), with a commonality that decreased physical activity appears with lockdowns. However, these studies are largely limited to current physical activity and mental health. To our knowledge, none have explored the possible association between prior parameters of physical fitness and pandemic-related depression.

From the existing evidence, it may be hypothesized that physical fitness factors are inversely correlated with depression during COVID-19. Therefore, this retrospective cross-sectional study's aims are two-fold: to assess the prevalence of depression in young adults during the pandemic and to investigate the associations between a variety of different measures of physical fitness, including prior anaerobic, aerobic, explosive, muscular, flexibility, and pulmonary fitness on the prevalence of depression during the COVID-19 pandemic.

# METHODS

## Study design and participants

This retrospective cross-sectional study enrolled two universities (Hunan Traditional Chinese Medical College, Hunan, China, and Medical College of Jinhua Polytechnic, Zhejiang, China) selected by convenience sampling, that performed the Chinese National Student Physical Fitness Standard (CNSPFS) battery between December 1st, 2019 to

January 20th, 2020 when government-issued sanctioned lockdowns and social distancing. A total of 14,059 university students who were free of chronic diseases and had completed a CNSPFS were screened. Of these, 13,013 participants (response rate of 93.2%) completed a follow-up questionnaire from May 1st to 23rd, 2020. Participants who provided poor quality questionnaires were excluded ($n$ = 124). The criteria of poor quality were: (1) If the ID information in the CNSPFS system did not match that of the follow-up questionnaire; or (2) If the 81-question survey was completed in less than 3 min. A total of 12,889 participants were included in the study. All baseline data were extracted from the CNSPFS system; the data during the COVID-19 pandemic were collected from the survey platform (https://www.wjx.cn). Data from the two time-points were linked by identifying each participant's university student ID number and matching them accordingly. This study was evaluated and approved by the Review Board of Xiangya Hospital Central South University (approval No. 202005126). Written informed consent was documented during the baseline, and digital informed consent was given upon initiating the survey. All participants were codified and anonymized to protect the confidentiality of individual participants.

## Physical fitness

Fitness measures were obtained from the completion of the CNSPFS battery, and scores were attained through the standardized scoring system that weighted each fitness indicator score by age- and sex-specific percentage. The CNSPFS battery included a 50-m sprint, (*Duffield, Dawson & Goodman, 2004*) an 800-m (women) and 1,000-m (men) run. *Margaria, Aghemo & Piñera Limas (1975)* a standing long jump (*Robertson & Fleming, 1987*) timed sit-ups (women) and pull-ups (men) (*Rutherford & Corbin, 1994*) sit and reach test and vital lung capacity respectively (*Ranu, Wilde & Madden, 2011*). Anaerobic, aerobic, explosive, muscular fitness, flexibility, and pulmonary fitness were assessed through these tests. The scores were classified as follows: low fitness (<60), moderate fitness (60–79), high fitness (80–89), and excellent fitness (>90). All tests were administered by trained physical education teachers following the CNSPFS standard operating procedures. The test-retest reliability across all assessments employed as an intraclass correlation coefficient (ICC) > 0.90. The details about performing CNSPFS have been described previously (*Yi et al., 2019*; *Zhu et al., 2017*).

## Baseline stress and depression during the COVID-19 pandemic

Baseline stress was assessed concurrently with the CNSPFS test through a modified question as described previously (*Frazier et al., 2014*). Beck's Depression Inventory, second edition (BDI-II) was also administered during the follow-up questionnaire performed in May (*Yang & Stewart, 2020*). The BDI-II is a 21-item self-report questionnaire validated in young Chinese adults (*Yang & Stewart, 2020*; *Zhu et al., 2018*) and well correlated with a clinical diagnosis of depression (*Moullec et al., 2015*) with four response options for each item. The added total scores of the BDI-II can vary from 0 to 63 and are classified as: 0–13 no depression, 14–19 mild depression, 20–28

moderate depression, and 29–63 severe depression (*Beck, Steer & Brown, 1996*; *Zhu et al., 2018*).

## Covariates

Exercise and physical activity habits were collected through a modified version with added items of the International Physical Activity Questionnaire-Long form-Chinese (IPAQ-LC) which had shown adequate reliability and reasonable validity for use in Chinese students (*Macfarlane, Chan & Cerin, 2011*). Questions regarded exercise pre-and during the lockdown, including frequency, duration, and intensity of aerobic exercise and strength exercise. Smoking status and alcohol intake were assessed as daily and weekly consumption, respectively. Socioeconomic status and dwelling location were retrieved from the university databases. Pre-pandemic body weight was assessed at the time of the CNSPFS test, while body weight during the COVID-19 pandemic was obtained via a self-reported questionnaire throughout May 2020 (*Ross, Eastman & Wing, 2019*).

## Characteristics of lockdown

The first lockdown order in China was delivered on January 23rd, 2020 by the government of Wuhan, Hubei province, followed by the other provinces across China. The main requirements were: (1) all individuals were ordered to stay home or at their place of residence, except for permitted work, local shopping, or other permitted errands, or as otherwise authorized. (2) All schools, sports facilities, entertainment, and recreational venues, personal care and beauty services, and the majority of factories and markets were closed.

## Statistical methods

For data of participants' demographics and characteristics, independent *t*-test and Chi-square test were used for assessment in mean difference between sexes of continuous and categorical variables, respectively.

The primary outcomes of the present study were the change in BDI-II depression scores (continuous variable) and the presence of depression (categorical variable), defined as a BDI-II score $\geq 14$ (*Beck, Steer & Brown, 1996*; *Zhu et al., 2018*) during the COVID-19 pandemic. The independent variables included prior fitness factors, stress, socioeconomic status, dwelling location, and smoking, alcohol, living status, changes in body weight, and exercise volume per week during the pandemic. The associations between each independent variable and BDI-II depression score change were assessed by univariate analyses. The independent relationships of prior physical fitness factors and BDI-II depression score change were assessed using multivariable linear regression.

The associations between prior physical fitness factors and the presence of depression, defined as a BDI-II depression score $\geq 14$, were evaluated via binary logistic regression in which the lowest level was set as the reference. The variables included in the regression models were demographics, the independent variables mentioned above and potentially associated with depression or those with a *p* value < 0.20 in the univariate analyses were included in the multivariable linear and binary logistic regression analyses.

To minimize confounding from baseline psychological status, baseline stress-stratified multivariable linear regression was performed. Analyses were carried out using SAS software, version 9.4 (SAS Institute), a two-tailed alpha level of 0.05 was considered to be statistically significant.

## RESULTS

### Demographics

Demographics of 12,889 participants are presented in Table 1. Male participants were proportionally a smaller percentage of the population. In the overall population, there were 12.2%, 60.6%, and 27.3% of participants that came from lower-income, middle-income, and upper-income families, respectively, and 35.4%, 42.4%, and 22.2% of participants lived in a rural, urban-rural junction, and urban areas, respectively. During the COVID-19 pandemic, 41% and 59% of participants lived alone or with family/friends, respectively.

### Characteristics of prior physical fitness and prevalence of depression during the COVID-19 pandemic

Each of five physical fitness parameters was scored and graded as low fitness (not pass), moderate fitness, good, or excellent according to CNSPFS that was established referring to national physical fitness test data while taking age and sex into account. Across multiple physical fitness tests, the mean score ranged from 53 to 78 in men and 67 to 78 in women, there were significant differences in physical fitness score between sexes ($P < 0.001$ for each comparison except pulmonary fitness score). The grade distributions were evaluated by the histogram and conformed to a normal distribution; the majority of participants were graded between moderate and high fitness. Less than 25% of participants were graded as low fitness or excellent. The median BDI-II depression during the pandemic scores were 2 for men and 1 for women ($P < 0.001$). Men and women who presented depression made up 13.9% and 15.0% of their respective populations, no significant difference was found between sexes ($P = 0.14$) (More in Table 1).

### Association between anaerobic fitness and depression during the pandemic

After multivariable adjustment, anaerobic fitness was independently and negatively associated with the BDI-II score. Across four grades, participants with excellent anaerobic fitness had an average BDI-II score that was −3.300 (95% CI [−4.761 to −1.838]) points lower than those with low fitness (Tables 2 and 3). This finding showed consistency in subjects with and without baseline stress (−2.689 [−4.199 to −1.178]) (Table 4). In binary (participants with depression vs without depression) logistic regression, compared to low fitness participants, participants with an excellent anaerobic fitness had less than half the risk of the presence of depression during the pandemic (OR 95% CI 0.58 [0.37–0.90]), and combined participants with moderate, high or excellent anaerobic fitness also showed a much lower risk (0.68 [0.55–0.82]) (Fig. 1).

**Table 1 Demographics and characteristics of participants pre- and during the COVID-19 pandemic.**

| | Men (N = 2,549) | Women (N = 10,340) | Total (N = 12,889) | Mean diff. (95% CI)* | P value* |
|---|---|---|---|---|---|
| Age, yr | 20 ± 1 | 19 ± 1 | 20 ± 1 | 0.2 [0.2–0.3] | <0.001 |
| Weight, kg | 63.4 ± 10.3 | 52.0 ± 7.7 | 54.3 ± 9.5 | 11.4 [11.0–11.8] | <0.001 |
| Body mass index, kg/m² | 21.2 ± 3.2 | 20.2 ± 2.7 | 20.4 ± 2.8 | 1.0 [0.8–1.1] | <0.001 |
| Socio-economic status† | | | | | 0.002 |
| Lower-income | 288 (11.3) | 1,282 (12.4) | 1,570 (12.2) | NA | |
| Middle-income | 1,621 (63.6) | 6,183 (59.8) | 7,804 (60.6) | NA | |
| Upper-income | 640 (25.1) | 2,875 (27.8) | 3,514 (27.3) | NA | |
| Baseline stress, yes, n (%) | 1,026 (40.3) | 4,458 (43.1) | 5,484 (42.5) | NA | 0.009 |
| Exercise volume, MET-hr/wk | 13.9 ± 12.3 | 8.8 ± 8.3 | 9.8 ± 9.4 | 5.2 [4.7–5.7] | <0.001 |
| Physical fitness | | | | | |
| Anaerobic fitness, 50-m sprint, s | 7.5 ± 0.5 | 9.2 ± 0.6 | 8.9 ± 0.9 | −1.7 [−1.7 to −1.7] | <0.001 |
| Anaerobic fitness score | 78 ± 10 | 70 ± 9 | 71 ± 10 | 8.4 [7.9–8.4] | <0.001 |
| Aerobic fitness, 800-m/1,000-m run, s | 237 ± 22 | 247 ± 27 | NA | NA | NA |
| Aerobic fitness score | 67 ± 13 | 73 ± 11 | 72 ± 12 | −5.8 [−6.3 to −5.2] | <0.001 |
| Explosive fitness, standing jump, meters | 2.2 ± 0.2 | 1.7 ± 0.1 | 1.8 ± 0.3 | 0.5 [0.4–0.6] | <0.001 |
| Explosive fitness score | 67 ± 14 | 71 ± 12 | 70 ± 13 | −3.6 [−4.3 to −3.0] | <0.001 |
| Muscular fitness, timed sit-ups/pull-ups | 34 ± 8 | 8 ± 5 | NA | NA | NA |
| Muscular fitness score | 53 ± 27 | 67 ± 11 | 65 ± 15 | −13.7 [−15.0 to −12.4] | <0.001 |
| Flexibility fitness, sit and reach, cm | 14 ± 6 | 17 ± 5 | 17 ± 6 | −3.7 [−3.9 to −3.4] | <0.001 |
| Flexibility fitness score | 73 ± 13 | 78 ± 11 | 77 ± 12 | −4.7 [−5.3 to −4.2] | <0.001 |
| Pulmonary fitness, vital capacity, L | 4.1 ± 0.7 | 2.8 ± 0.4 | 3.1 ± 0.7 | 1.3 [1.3–1.3] | <0.001 |
| Pulmonary fitness score | 75 ± 13 | 76 ± 11 | 76 ± 11 | −0.5 [−1.0 to 0.1] | 0.100 |
| Depression score during COVID-19 pandemic | 5 ± 8 | 6 ± 8 | 6 ± 8 | −0.6 [−0.9 to −0.2] | 0.001 |
| Without depression | 2,195 (86.1) | 8,789 (85.0) | 10,984 (85.2) | NA | 0.151 |
| With depression | 354 (13.9) | 1,551 (15.0) | 1,905 (14.8) | NA | |
| Mild depression | 155 (6.1) | 672 (6.5) | 827 (6.4) | NA | 0.650 |
| Moderate depression | 138 (5.4) | 579 (5.6) | 717 (5.6) | NA | |
| Severe depression | 61 (2.4) | 300 (2.9) | 361 (2.8) | NA | |
| Geographic attribute | | | | | <0.001 |
| Rural area | 1,007 (39.5) | 3,557 (34.4) | 4,564 (35.4) | NA | |
| Urban-rural junction area | 1,037 (40.7) | 4,426 (42.8) | 5,463 (42.4) | NA | |
| Urban area | 505 (19.8) | 2,358 (22.8) | 2,862 (22.2) | NA | |
| Living status during COVID-19 pandemic | | | | | <0.001 |
| Living alone | 1,207 (47.4) | 4,081 (39.5) | 5,288 (41.0) | NA | |
| Living with family/friends | 1,342 (52.6) | 6,259 (60.5) | 7,601 (59.0) | NA | |
| Smoking, n (%) | | | | | <0.001 |
| Never smoke | 1932 (75.8) | 10,107 (97.7) | 12,039 (93.4) | NA | |
| Former smoker | 227 (8.9) | 147 (1.4) | 374 (2.9) | NA | |
| Currently smoking | 390 (15.3) | 86 (0.9) | 476 (3.7) | NA | |
| <10 cigarets/day | 297 (11.7) | 73 (0.7) | 370 (2.9) | NA | |
| 10–15 cigarets/day | 57 (2.2) | 9 (0.09) | 66 (0.5) | NA | |

| | Men (N = 2,549) | Women (N = 10,340) | Total (N = 12,889) | Mean diff. (95% CI)* | P value* |
|---|---|---|---|---|---|
| Table 1 (continued) | | | | | |
| >15 cigarets/day | 36 (1.4) | 4 (0.03) | 40 (0.3) | NA | |
| Alcohol, drinks/wk | 1.8 ± 2.9 | 0.7 ± 1.6 | 0.9 ± 2.0 | 1.2 [0.1–1.3] | <0.001 |
| Change in weight, kg | 2.6 ± 3.9 | 2.1 ± 3.6 | 2.2 ± 3.7 | 0.5 [0.3–0.6] | <0.001 |
| Change in exercise volume, MET-hr/wk | −0.6 ± 13.0 | 2.5 ± 10.1 | 1.9 ± 10.8 | −3.1 [−3.6 to −2.5] | <0.001 |

**Notes:**
† Socio-economic level data was obtained based on residence place of participants and 2019 Chinese Family Income data. A family earning less than ¥14,360 per year was considered a lower-income family; between ¥14,360 and ¥36,470 per year was considered a middle-income family; more than ¥36,470 per year was considered an upper-income family.
* Mean difference was calculated as women subtracted by men, and expressed as mean difference (95% CI). P values are for the comparison between men and women. CI, confidence interval; COVID-19, coronavirus disease-19. Data were expressed as mean ± standard deviation or number (percent) accordingly.

## Association between aerobic fitness and depression during the pandemic

Aerobic fitness showed a similar association with BDI-II score. Across four grades of aerobic fitness, participants categorized as excellent had an average BDI-II score on average −1.521 (95% CI [−2.586 to −0.455]) lower than those with low fitness (Table 3). This finding was consistent in participants with and without baseline stress (−1.772 [−2.846 to −0.699]) (Table 4). In binary logistic regression, compared to those who had low fitness scores, combined participants with moderate, high, or excellent aerobic fitness grades showed a lower risk of the presence of depression (OR 95% CI 0.80 [0.68–0.91]) (Fig. 1).

## Association between explosive fitness and depression during the pandemic

Explosive fitness was inversely correlated with depression during the COVID-19 pandemic. Participants with excellent explosive fitness had an average of −1.643 (95% CI [−92.673 to −0.613]) lower BDI-II score than participants with low fitness (Tables 2 and 3). This finding was consistent in participants with and without baseline stress across sex (−1.849 [−2.884 to −0.815]) (Table 4). In binary logistic regression, compared to those with low fitness scores, participants with excellent explosive fitness had a 36% lower risk of the presence of depression (OR 95% CI 0.62 [0.46–0.85]), and combined participants with moderate, high or excellent explosive fitness scores also showed a lower risk (0.72 [0.61–0.82]) (Fig. 1).

## Association between muscular fitness and depression during the pandemic

Muscular fitness was also independently and negatively correlated with depression. Participants with excellent muscular fitness had an average of −1.713 (95% CI [−2.956 to −0.470]) lower BDI-II score than participants with low fitness (Table 3). This finding was consistent in those with and without baseline stress (−2.136 [−3.383 to −0.889]) (Table 4). In binary logistic regression, compared to those who were categorized as low fitness, participants with excellent muscular fitness had less than half risk of the presence of

**Table 2 Univariate linear regression for the relationships between prior physical fitness and depression during the COVID-19 pandemic.**

| | Men (N = 2,549) | | Women (N = 10,340) | | Total (N = 12,889) | |
|---|---|---|---|---|---|---|
| | Coefficients (95% CI) | P Value | Coefficients (95% CI) | P Value | Coefficients (95% CI) | P Value |
| Age, yr | −0.100 [−0.352 to 0.151] | 0.432 | −0.038 [−0.172 to 0.095] | 0.574 | −0.064 [−0.182 to 0.054] | 0.287 |
| Change in weight, kg | 0.147 [0.066–0.228] | <0.001 | 0.127 [0.082–0.171] | <0.001 | 0.128 [0.089–0.167] | <0.001 |
| Change in exercise volume, MET-hr/wk | −0.010 [−0.035 to 0.014] | 0.411 | −0.021 [−0.037 to −0.005] | 0.010 | −0.015 [−0.029 to −0.002] | 0.025 |
| Baseline stress, yes | | | | | | |
| No | Reference | | Reference | | Reference | |
| Yes | 0.467 [0.144–0.789] | 0.005 | 0.283 [0.120–0.446] | <0.001 | 0.324 [0.179–0.470] | <0.001 |
| Prior physical fitness | | | | | | |
| Anaerobic fitness | | | | | | |
| Low | Reference | | Reference | | Reference | |
| Moderate | −3.471 [−7.368 to 0.426] | 0.084 | −1.504 [−2.502 to −0.505] | 0.001 | −1.635 [−2.580 to −0.690] | <0.001 |
| High | −3.929 [−7.919 to 0.060] | 0.054 | −1.504 [−0.706 to −0.302] | 0.010 | −1.816 [−2.906 to −0.727] | <0.001 |
| Excellent | −4.547 [−8.504 to −0.590] | 0.023 | −2.680 [−4.716 to −0.443] | 0.014 | −2.964 [−4.197 to −1.732] | <0.001 |
| Aerobic fitness | | | | | | |
| Low | Reference | | Reference | | Reference | |
| Moderate | −1.123 [−2.176 to −0.070] | 0.033 | −1.497 [−2.322 to −0.673] | <0.001 | −1.120 [−1.841 to −0.558] | <0.001 |
| High | −0.978 [−2.550 to 0.594] | 0.340 | −1.548 [−2.460 to −0.637] | <0.001 | −1.176 [−1.919 to −0.434] | <0.001 |
| Excellent | −2.648 [−4.887 to −0.408] | 0.015 | −2.017 [−3.103 to −0.931] | <0.001 | −1.794 [−2.723 to −0.865] | <0.001 |
| Explosive fitness | | | | | | |
| Low | Reference | | Reference | | Reference | |
| Moderate | −1.236 [−2.452 to −0.019] | 0.046 | −0.652 [−1.399 to 0.100] | 0.100 | −0.745 [−1.383 to −0.108] | 0.017 |
| | | 0.276 | −0.720 [−1.602 to 0.162] | 0.132 | −0.759 [−1.523 to 0.005] | 0.052 |
| Excellent | −2.486 [−4.765 to −0.208] | 0.028 | −1.531 [−2.545 to −0.517] | 0.001 | −1.605 [−2.513 to −0.697] | <0.001 |
| Muscular fitness | | | | | | |
| Low | Reference | | Reference | | Reference | |
| Moderate | 0.434 [−0.316 to 1.184] | 0.263 | −1.410 [−2.100 to −0.721) | <0.001 | -0.139 [-0.586 to 0.307] | 0.824 |
| High | 0.152 [−1.265 to 1.569] | 0.860 | −2.438 [−3.476 to −1.400] | <0.001 | −1.044 [−1.871 to −0.218] | 0.008 |
| Excellent | −1.610 [−2.952 to −0.268] | 0.018 | −2.152 [−3.618 to −0.687) | 0.016 | −1.482 [−2.562 to −0.402] | 0.003 |
| Flexibility fitness | | | | | | |
| Low | Reference | | Reference | | Reference | |
| Moderate | 0.253 [−1.350 to 1.856] | 0.946 | −0.479 [−2.009 to 1.051) | 0.627 | 0.061 [−1.031 to 1.154] | 0.994 |
| High | −0.759 [−2.593 to 1.074] | 0.572 | −0.804 [−2.364 to 0.756] | 0.358 | −0.304 [−1.436 to 0.827] | 0.726 |
| Excellent | 0.549 [−1.336 to 2.433] | 0.772 | −0.763 [−2.324 to 0.799] | 0.389 | −0.088 [−1.223 to 1.047] | 0.984 |
| Pulmonary fitness | | | | | | |
| Low | Reference | | Reference | | Reference | |
| Moderate | −0.769 [−2.562 to 1.024] | 0.520 | 0.783 [−0.439 to 2.001] | 0.248 | 0.380 [−0.629 to 1.391] | 0.580 |
| High | −1.196 [−3.107 to 0.716] | 0.272 | 0.599 [−0.691 to 1.890] | 0.448 | 0.089 [−0.981 to 1.160] | 0.985 |
| Excellent | −0.749 [−2.725 to 1.227] | 0.596 | 0.177 [−1.136 to 1.490] | 0.942 | −0.118 [−1.211 to 0.975] | 0.970 |

| | Men (N = 2,549) | | Women (N = 10,340) | | Total (N = 12,889) | |
|---|---|---|---|---|---|---|
| | Coefficients (95% CI) | P Value | Coefficients (95% CI) | P Value | Coefficients (95% CI) | P Value |
| Geographic attribute | | | | | | |
| Rural area | Reference | | Reference | | Reference | |
| Urban-rural junction area | −0.941 [−2.149 to 0.268] | 0.147 | 0.063 [0.484 to 0.609] | 0.948 | −0.115 [−0.612 to 0.383] | 0.813 |
| Urban area | −1.232 [−2.491 to 0.027] | 0.056 | −0.202 [−0.751 to 0.348] | 0.605 | −0.378 [−0.881 to 0.125] | 0.164 |
| Socio-economic status | | | | | | |
| Lower-income | Reference | | Reference | | Reference | |
| Middle-income | −0.488 [−1.934 to 0.957] | 0.631 | −0.279 [−0.917 to 0.358] | 0.482 | −0.310 [−0.893 to 0.273] | 0.362 |
| Upper-income | −0.261 [−1.843 to 1.320] | 0.886 | −1.163 [−1.870 to −0.455] | <0.001 | −1.001 [−1.652 to −0.361] | 0.001 |
| Alcohol, drinks/wk | 0.422 [0.312–0.532] | <0.001 | 0.766 [0.666–0.866] | <0.001 | 0.554 [0.481–0.626] | <0.001 |
| Living status | | | | | | |
| Living alone | Reference | | Reference | | Reference | |
| Living with family/friends | 0.222 [−0.539 to 0.096] | 0.171 | 0.068 [−0.233 to 0.097] | 0.419 | 0.114 [−0.261 to 0.032] | 0.126 |
| Smoking | | | | | | |
| Never smoke | Reference | | Reference | | Reference | |
| Former smoker | 2.270 [1.017–3.523] | <0.001 | 6.787 [5.402–8.173] | <0.001 | 3.610 [2.736–4.483] | <0.001 |
| Currently smoking | | | | | | |
| <10 cigarets/day | 1.238 [0.124–2.353] | 0.027 | 9.185 [7.232–11.137] | <0.001 | 2.196 [1.318–3.074] | <0.001 |
| 10-15 cigarets/day | 5.327 [2.927–7.727] | <0.001 | 9.669 [4.126–15.212] | <0.001 | 5.246 [3.195–7.298] | <0.001 |
| >15 cigarets/day | 5.989 [2.854–9.124] | <0.001 | 12.835 [4.523–21.148] | 0.002 | 6.029 [3.293–8.766] | <0.001 |

**Note:**
CI, confidence interval; COVID-19, coronavirus disease-19; BDI-II, Becks' Depression Inventory-II. The first level of the ordered categorical variables were used as reference. Data are expressed as BDI-II score change related to each variable.

depression during the pandemic (OR 95% CI 0.47 [0.31 to 0.72]), and combined participants with moderate, high or excellent muscular fitness also showed a much lower risk (0.66 [0.57 to 0.75]) (Fig. 1).

# DISCUSSION

This study is the first to examine the relationship between physical fitness across multiple domains and the presence of depressive symptoms during the COVID-19 pandemic. Our data showed that 13.9% of young men and 15.0% of women had BDI scores that qualified them for the diagnosis of depression, which is unexpectedly similar to the level of depression reported previously in older adult populations (16.5%) (Wang et al., 2020). Findings that are particularly implicative as both depression and low physical fitness are risk factors for future CVD and diabetes (Moulton, Pickup & Ismail, 2015; Myers et al., 2015; Pedersen et al., 2019). This study further explored the association between previous levels of explosive, anaerobic, flexibility, and pulmonary domains of fitness, with depression. Our data demonstrate that apart from flexibility and pulmonary fitness, all of these fitness parameters were independently inversely associated with depression in young adults free from chronic diseases.

**Table 3 Multivariate linear regression for the relationships between prior physical fitness and depression during the COVID-19 pandemic.**

| | Men (N = 2,549) | | Women (N = 10,340) | | Total (N = 12,889) | |
|---|---|---|---|---|---|---|
| | Coefficient (95% CI) | P Value | Coefficient (95% CI) | P Value | Coefficient (95% CI) | P Value |
| **Anaerobic fitness** | | | | | | |
| Low | Reference | | Reference | | Reference | |
| Moderate | −1.564 [−6.263 to 3.135] | 0.292 | −1.544 [−2.639 to −0.450] | 0.003 | −1.612 [−2.668 to −0.557] | 0.001 |
| High | −2.336 [−7.156 to 2.484] | 0.390 | −1.558 [−2.878 to −0.238] | 0.016 | −1.958 [−3.168 to −0.731] | <0.001 |
| Excellent | −2.593 [−7.375 to 2.187] | 0.325 | −2.712 [−5.192 to −0.233] | 0.028 | −3.300 [−4.761 to −1.838] | <0.001 |
| **Aerobic fitness** | | | | | | |
| Low | Reference | | Reference | | Reference | |
| Moderate | −0.923 [−2.306 to 0.460] | 0.279 | −1.449 [−2.376 to −0.522] | 0.001 | −1.137 [−1.896 to −0.378] | <0.002 |
| High | −0.913 [−2.966 to 1.140] | 0.605 | −1.447 [−2.466 to −0.429] | 0.003 | −1.071 [−1.935 to −0.207] | <0.011 |
| Excellent | −2.152 [−5.101 to 0.797] | 0.212 | −1.820 [−3.026 to −0.615] | 0.002 | −1.521 [−2.586 to −0.455] | 0.003 |
| **Explosive fitness** | | | | | | |
| Low | Reference | | Reference | | Reference | |
| Moderate | −1.797 [−3.326 to −0.268] | 0.016 | −0.693 [−1.536 to 0.150] | 0.127 | −0.851 [−1.587 to −0.115] | 0.019 |
| High | −1.355 [−3.327 to 0.616] | 0.250 | −0.859 [−1.844 to 0.126] | 0.100 | −0.923 [−1.798 to −0.048] | 0.036 |
| Excellent | −3.139 [−6.441 to 0.164] | 0.067 | −1.550 [−2.669 to −0.431] | 0.004 | −1.643 [−2.673 to −0.613] | <0.001 |
| **Muscular fitness** | | | | | | |
| Low | Reference | | Reference | | Reference | |
| Moderate | −0.777 [−1.919 to 0.364] | 0.277 | −1.601 [−2.379 to −0.823] | <0.001 | −0.498 [−1.049 to 0.053] | 0.088 |
| High | −1.408 [−3.560 to 0.744] | 0.310 | −2.324 [−3.482 to −1.166] | <0.001 | −1.355 [−2.3 to −0.398] | 0.002 |
| Excellent | −2.171 [−4.207 to −0.135] | 0.033 | −2.084 [−3.680 to −0.488] | 0.006 | −1.713 [2.956 to −0.470] | 0.003 |

**Note:**
CI, confidence interval; COVID-19, coronavirus disease-19; BDI-II, Becks' Depression Inventory-II. Multivariate models were adjusted for age, baseline stress, dwelling location, socio-economic level, and smoking, alcohol, living status, changes in weight and exercise volume during COVID-19 pandemic. The low fitness level of each physical fitness variable were used as reference. Data are expressed as BDI-II score change related to each variable.

Depression and depressive symptom severity vary in individuals, and at different periods, at times, it may impact functional capabilities, whose effects are deleterious on individuals' physical and psychological wellbeing. While the etiology of depression remains largely unknown, the onset of depression may come in various forms, ranging from physiological factors, psychological factors, and those coming from an individual's environment. Both chronic and acute stresses are believed to play an integral part in the development of depression. Chronic stress, encompassing long-term negative environmental circumstances, such as financial difficulties; conflictual relationships with family, friends, or romantic partners (*Hammen et al., 2009*), acute stresses coming from episodes of stress such as those environmental or from personal loss. The COVID-19 pandemic and lockdowns have led to a surge in negative emotions and increased depression (*Choi, Hui & Wan, 2020*; *Lei et al., 2020*).

Recent research has shown that anaerobic training to be inversely associated with depression severity independent of aerobic activity (*Cangin et al., 2018*). Our research showed for the first time that anaerobic fitness was the greatest predictor of a lower BDI-II score among the fitness parameters. After stratification of baseline stress, increasing levels

**Table 4 Baseline stress-stratified multivariate linear regression for the relationships between prior physical fitness and depression during the COVID-19 pandemic.**

| | No baseline stress (N = 2,549) | | Baseline stress (N = 10,340) | | Total (N = 12,889) | |
|---|---|---|---|---|---|---|
| | Coefficient (95% CI) | P Value | Coefficient (95% CI) | P Value | Coefficient (95% CI) | P Value |
| **Anaerobic fitness** | | | | | | |
| Low | Reference | | Reference | | Reference | |
| Moderate | −1.501 [−2.843 to −0.158] | 0.025 | −1.613 [−3.309 to 0.082] | 0.065 | −1.543 [−2.601 to −0.486] | 0.002 |
| High | −2.101 [−3.679 to −0.523] | 0.006 | −1.482 [−3.442 to 0.479] | 0.172 | −1.762 [−2.995 to −0.528] | 0.003 |
| Excellent | −2.628 [−4.518 to −0.738] | 0.004 | −2.821 [−5.298 to −0.343] | 0.022 | −2.689 [−4.199 to −1.178] | <0.001 |
| **Aerobic fitness** | | | | | | |
| Low | Reference | | Reference | | Reference | |
| Moderate | −1.020 [−1.953 to −0.087] | 0.029 | −1.976 [−3.276 to −0.676] | 0.001 | −1.346 [−2.111 to −0.581] | <0.001 |
| High | −1.257 [−2.338 to −0.175] | 0.018 | −1.700 [−3.159 to −0.242] | 0.018 | −1.340 [−2.214 to −0.466] | 0.001 |
| Excellent | −1.579 [−2.947 to −0.211] | 0.019 | −2.380 [−4.111 to −0.649] | 0.004 | −1.772 [−2.846 to −0.699] | <0.001 |
| **Explosive fitness** | | | | | | |
| Low | Reference | | Reference | | Reference | |
| Moderate | −0.836 [−1.759 to 0.087] | 0.084 | −1.197 [−2.404 to 0.012] | 0.053 | 0.983 [−1.721 to −0.245] | 0.006 |
| High | −0.502 [−1.604 to 0.600] | 0.538 | −1.786 [−3.212 to −0.360] | 0.010 | −1.040 [−1.916 to −0.163] | 0.016 |
| Excellent | −1.448 [−2.763 to −0.133] | 0.027 | −2.456 [−4.114 to −0.798] | 0.002 | −1.849 [−2.884 to −0.815] | <0.001 |
| **Muscular fitness** | | | | | | |
| Low | Reference | | Reference | | Reference | |
| Moderate | −1.095 [−1.884 to −0.305] | 0.003 | −1.903 [−2.927 to −0.879] | <0.001 | −1.419 [−2.048 to −0.792] | <0.001 |
| High | −1.568 [−2.774 to −0.362] | 0.006 | −2.963 [−4.634 to −1.292] | <0.001 | −2.139 [−3.125 to −1.153] | <0.001 |
| Excellent | −2.399 [−3.958 to −0.841] | <0.001 | −1.749 [−3.723 to 0.296] | 0.115 | −2.136 [−3.383 to −0.889] | <0.001 |

Note:
CI, confidence interval; COVID-19, coronavirus disease-19; BDI-II, Becks' Depression Inventory-II. Multivariate models were adjusted for sex, age, dwelling location, socio-economic level, and smoking, alcohol, living status, changes in weight and exercise volume during COVID-19 pandemic. The low fitness level of each physical fitness variable were used as reference. Data are expressed as BDI-II score change related to each variable.

of fitness were significantly correlated with lower BDI-II scores in a near dose-response fashion, with those with the highest fitness being less likely to be depressed (Table 4). Furthermore, in conjunction with previous research (*Harvey et al., 2018*; *Kerling et al., 2015*) our findings showed that aerobic fitness was independently associated with lower depression scores. A number of studies have found that aerobic exercise is an effective therapy for moderate forms of depression while also showing it may be equally effective compared to other traditional methods of psychotherapy. The association of aerobic fitness with lower levels of depression is not limited to young adults as longitudinal studies have similar results in reducing depression in middle-aged and older individuals and younger adults (*Jaworska et al., 2019*).

Muscular fitness was measured through two strength tests to best evaluate for muscular strength and endurance, given the variability of muscular fitness across sexes. Muscular fitness and its inverse association with depressive symptoms have been previously investigated (*Marques et al., 2020*). This association was also shown in this study after multivariable analysis, showing that participants in the highest subset of muscular fitness

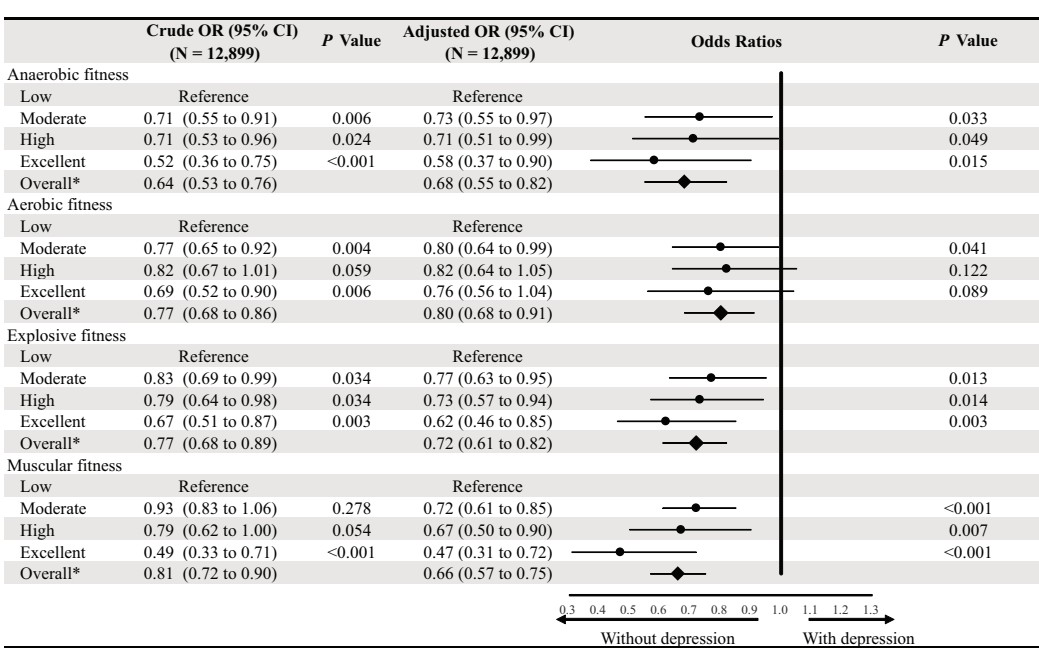

| | Crude OR (95% CI) (N = 12,899) | P Value | Adjusted OR (95% CI) (N = 12,899) | Odds Ratios | P Value |
|---|---|---|---|---|---|
| **Anaerobic fitness** | | | | | |
| Low | Reference | | Reference | | |
| Moderate | 0.71 (0.55 to 0.91) | 0.006 | 0.73 (0.55 to 0.97) | | 0.033 |
| High | 0.71 (0.53 to 0.96) | 0.024 | 0.71 (0.51 to 0.99) | | 0.049 |
| Excellent | 0.52 (0.36 to 0.75) | <0.001 | 0.58 (0.37 to 0.90) | | 0.015 |
| Overall* | 0.64 (0.53 to 0.76) | | 0.68 (0.55 to 0.82) | | |
| **Aerobic fitness** | | | | | |
| Low | Reference | | Reference | | |
| Moderate | 0.77 (0.65 to 0.92) | 0.004 | 0.80 (0.64 to 0.99) | | 0.041 |
| High | 0.82 (0.67 to 1.01) | 0.059 | 0.82 (0.64 to 1.05) | | 0.122 |
| Excellent | 0.69 (0.52 to 0.90) | 0.006 | 0.76 (0.56 to 1.04) | | 0.089 |
| Overall* | 0.77 (0.68 to 0.86) | | 0.80 (0.68 to 0.91) | | |
| **Explosive fitness** | | | | | |
| Low | Reference | | Reference | | |
| Moderate | 0.83 (0.69 to 0.99) | 0.034 | 0.77 (0.63 to 0.95) | | 0.013 |
| High | 0.79 (0.64 to 0.98) | 0.034 | 0.73 (0.57 to 0.94) | | 0.014 |
| Excellent | 0.67 (0.51 to 0.87) | 0.003 | 0.62 (0.46 to 0.85) | | 0.003 |
| Overall* | 0.77 (0.68 to 0.89) | | 0.72 (0.61 to 0.82) | | |
| **Muscular fitness** | | | | | |
| Low | Reference | | Reference | | |
| Moderate | 0.93 (0.83 to 1.06) | 0.278 | 0.72 (0.61 to 0.85) | | <0.001 |
| High | 0.79 (0.62 to 1.00) | 0.054 | 0.67 (0.50 to 0.90) | | 0.007 |
| Excellent | 0.49 (0.33 to 0.71) | <0.001 | 0.47 (0.31 to 0.72) | | <0.001 |
| Overall* | 0.81 (0.72 to 0.90) | | 0.66 (0.57 to 0.75) | | |

0.3 0.4 0.5 0.6 0.7 0.8 0.9 1.0 1.1 1.2 1.3
Without depression — With depression

**Figure 1 Binary logistic regression for the relationships between prior physical fitness and depression during the COVID-19 pandemic.** OR, odds ratio; CI, confidence interval; COVID-19, coronavirus disease-19. The low fitness was used as reference for each physical fitness variable. Multivariate models were adjusted for sex, age, prior perceived stress, dwelling location, socio-economic level, and smoking, alcohol, living status, changes in weight and exercise volume during the COVID-19 pandemic. *Overall equates to the participants that were graded as pass, good and excellent.

demonstrated a BDI-II score lower than participants with low fitness scores. These findings are consistent with those found during pre-pandemic conditions, where increased handgrip strength as well as comprehensive muscular fitness appears to act as a buffer against depressive symptoms and are associated with improvements in depression (*Krogh et al., 2009*; *Suija et al., 2013*). Previous large-scale observational studies suggested that muscular fitness may even have greater effects than those associated with aerobic fitness in the reduction of depression (*Bennie et al., 2019*). Our findings extend those of others, in that we found muscular fitness to be associated with lower depression scores during the COVID-19 pandemic. Moreover, Explosive fitness, utilizing the phosphocreatine system (*Baker, McCormick & Robergs, 2010*), was found to be independently and inversely associated with depressive symptoms, with the highest level of explosive fitness being associated with a lower BDI-II score. Furthermore, individuals in the highest explosive fitness level had less than half the associated risk of depression than individuals in the lowest levels. This is a particularly interesting finding as it appears to suggest that different types of fitness than those reported elsewhere (*Ren et al., 2020*) may have an effect of protection from depressive symptoms. Future prospective cohort or intervention studies are required to better elucidate the relationships between explosive fitness and the risk of depressive symptoms.

Physical fitness and its inverse relationship with depression could be brought about through increased physical activity and exercise. Accumulating evidence suggests that

exercise could be associated with primary monoamines, whereby higher amounts of exercise have a positive impact on neurotransmitter systems that regulate primary monoamines, dopamine, noradrenaline, and serotonin (*Dishman, 1997*; *Poulton & Muir, 2005*). As well as having a positive effect on physiological states, regular exercise and higher fitness could impact psychological states. Mental resilience may be improved through exercise and attainability of higher fitness (*Childs & De Wit, 2014*) as persons who feel or believe themselves to be healthier may be less inclined to fear the COVID-19 pandemic or at least feel more resilient against it. This has been demonstrated as those with higher mental resilience were less likely to be depressed (*Kirby et al., 2017*) as participation in regular exercise can usually distract individuals from noxious stimuli, thereby improving depression. Furthermore, since exercise is often extrinsic, increased self-efficacy and self-esteem, garnered from higher muscle mass and fitness could lead to improved mental health (*Blumenthal et al., 2007*).

While this study provides certain insights into the associations through which higher physical fitness levels in several parameters of fitness is related to a decreased risk of depressive symptoms, the causation between fitness parameters and depressive symptoms could not be established due to the cross-sectional study design. Associations between fitness measures and depressive symptoms in young adults may be bidirectional. Studies have indicated that previous depression could be a factor in the cessation of exercise, and the development of a sedentary lifestyle since those with depression may have lower self-worth, and confidence, increased self-criticism, and unwarranted guilt (*Faulkner, Carson & Stone, 2014*; *Ren et al., 2020*). Future prospective cohort or intervention studies are required to better elucidate the relationships between fitness, particularly anaerobic and explosive fitness, and the risk of depressive symptoms.

Furthermore, the sample was selected through convenience sampling and therefore may be subject to a degree of information bias, may not be representative of the whole population, and variations within individuals exist. As is with all cross-sectional studies, our data demonstrate an association between physical fitness and prevalence of rather than incidence of depression during the COVID-19 pandemic, although baseline stress status was included as a covariate in all multivariable analyses, and baseline stress-stratified multivariable analysis was performed, with consistent findings. The present study only comprised university-educated Chinese young adults, which potentially limits the generalizability of the findings. However, our findings were consistent among sexes, and with multivariable adjustment of age, baseline stress, geographic attribute, socio-economic level, and smoking, alcohol, living status, changes in weight, and exercise volume during the pandemic. In addition, all findings were consistent in both multivariable linear and binary logistic regressions. It seems plausible that the biologic effects of many factors would be qualitatively similar in other populations.

## CONCLUSION

Multiple physical fitness parameters are inversely associated with the prevalence of depression in young adults during the COVID-19 pandemic. These associations were independent of other potential confounders, such as sex, age, baseline stress, dwelling

location, socio-economic level, and smoking, alcohol, living status, weight changes, and exercise volume during the pandemic. These findings along with previous research suggest that that techniques and lifestyle management that lead to improved comprehensive fitness including a wider range of muscle groups and energy systems may be considered as a possible approach to help prevent and/or reverse depression during the pandemic.

## ACKNOWLEDGEMENTS

We would like to thank all participants and investigators in participating centers.

### Funding

This study is supported by Hunan Development and Reform Commission Foundation of China (No. (2012)1521 to Suixin Liu) and The Youth Science Foundation of Xiangya Hospital (No. 2019Q03 to Yaoshan Dun). The funders had no role in study design, data collection and analysis, decision to publish, or preparation of the manuscript.

### Grant Disclosures

The following grant information was disclosed by the authors:
Hunan Development and Reform Commission Foundation of China: (2012)1521.
Youth Science Foundation of Xiangya Hospital: 2019Q03.

### Competing Interests

The authors declare that they have no competing interests.

### Author Contributions

- Yaoshan Dun conceived and designed the experiments, performed the experiments, analyzed the data, prepared figures and/or tables, authored or reviewed drafts of the paper, and approved the final draft.
- Jeffrey W. Ripley-Gonzalez conceived and designed the experiments, performed the experiments, analyzed the data, prepared figures and/or tables, authored or reviewed drafts of the paper, and approved the final draft.
- Nanjiang Zhou analyzed the data, prepared figures and/or tables, coordinated the study; conduct and collect data, and approved the final draft.
- Qiuxia Li performed the experiments, prepared figures and/or tables, authored or reviewed drafts of the paper, coordinated the study; conduct and collect data, and approved the final draft.
- Meijuan Chen analyzed the data, prepared figures and/or tables, and approved the final draft.
- Zihang Hu performed the experiments, prepared figures and/or tables, coordinated the study; conduct and collect data, and approved the final draft.
- Wenliang Zhang analyzed the data, prepared figures and/or tables, coordinated the study; conduct and collect data, and approved the final draft.

- Randal J. Thomas performed the experiments, authored or reviewed drafts of the paper, and approved the final draft.
- Thomas P. Olson performed the experiments, authored or reviewed drafts of the paper, and approved the final draft.
- Jie Liu analyzed the data, prepared figures and/or tables, coordinated the study; conduct and collect data, and approved the final draft.
- Yuchen Dong analyzed the data, prepared figures and/or tables, coordinated the study; conduct and collect data, and approved the final draft.
- Suixin Liu conceived and designed the experiments, prepared figures and/or tables, authored or reviewed drafts of the paper, and approved the final draft.

## Human Ethics

The following information was supplied relating to ethical approvals (i.e., approving body and any reference numbers):

The study was approved by Ethics Committee of Xiangya Hospital of Central South University (approval no. 202005126) and written informed consent was documented. All participants were codified and anonymized to protect the confidentiality of individual participants.

## Data Availability

Raw data, including the data of anonymized participants fitness scores, anthropometric measurements and psychological status, are available as a Supplemental File.

## Supplemental Information

Supplemental information for this article can be found online at http://dx.doi.org/10.7717/peerj.11091#supplemental-information.

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
