# Peer review of "The association between prior physical fitness and depression in young adults during the COVID-19 pandemic—a cross-sectional, retrospective study"

_PeerJ, doi:10.7717/peerj.11091_

## Round 0.1 · original submission · Major Revisions

Some experts have reviewed your work and have found scientific merit in your methods. However, there are some issues that you should address in a revised version of the text.

Reviewer 1 ·

Basic reporting

This is an original manuscript with no main problems at all. My main concern is the confusion when no detailing OR values.

Experimental design

This is an observational study.

Validity of the findings

The findings are interesting. They add something new since there is nothing related to physical fitness during the pandemic.

Additional comments

This is an interesting manuscript dealing with a hot topic. Indeed, little is known about physical fitness assessments associated to mental health in the Covid-19 pandemic. However, there are still several points that should be addressed.
Abstract
-Indicate percentage of women and mean age of participants
-Why don´t you report ORs for fitter individuals instead of a p value
Introduction
-Lines 102-105. There are prior research showing inverse association between mental health and physical activity during the Covid-19 pandemic, which should be commented, at least, in the introduction. Otherwise, it seems as there has been no revision of the current literature on the topic:
https://www.frontiersin.org/articles/10.3389/fpsyt.2020.00729/full
Later, you can add the comment that your study is the first examining the association between physical fitness and mental health during the pandemic.
-You have stated no hypothesis
Methods
-Some words stating that the sample has been selected by convenience sampling.
-How did you link information of the questionnaire with information from the dataset (fitness)
-I am missing the point in statistics. If you conducted binary logistic regression, you got ORs as a result.
Results
-Lines 196-197. I suggest avoiding interpretations in the results section.
-Line 220. Points or ORs?
Discussion
-Since your sample is not representative, it should be stated the possibility of an information bias.
Tables
-Table 2 values are ORs. Aren´t they? You should indicate this.

Reviewer 2 ·

Basic reporting

The manuscript is generally well written and clear. It is well structured with no fatal flaws. generally is it well references though I question the use of some references in my other comments

Experimental design

The design is generally consistent with the aims. The knowledge gap this study fills is understaed and more could be done to highlight the significance of the study, especially in the introduction

Validity of the findings

All relevant data appear to be provided, however a statistical review is recommended. Some conclusions are overstated and may required revising.

Additional comments

Overall comments
This is a useful study that differs in important ways from other covid-related works. There are some suggestions requiring attention.

Abstract
Conclusions. The second point that improving physical fitness may represent an effective approach to help prevent/reverse depression cannot be confirmed from the available data and should be revised.

L91. This sentence is very long, contains too many elements and should be revised.
L103. Sample studies should be cited on the association between COVID-19 and changes in mental health, being cautions that the timing of data collection, population sampled, social restrictions and other factors vary between studies.

L106. The introduction does not provide a basis to address the first aim of the study, as there are multiple papers already on this topic, and stated limitations of previous work is the single factor approach to fitness. The logic link missing in this introduction is most studies have examined the association between exercise and / or physical activity participation and mental health during the pandemic , NOT fitness and mental health during the pandemic. This latter point is the real novelty of this study, and has not been sufficiently emphasised.

L127. Please explain how written informed consent was provided when data were collected using an online survey, and how data were match between timepoints.

L146. It is not clear if BDI and stress where administered only at baseline, or during May, presumably using the online survey system. Please clarify.

L153. Please provide a reference for the cutpoints for depression severity scores..

L156. Please explain what tools were used to measure health behaviours (exercise/PA, smoking, etc) including the validity and reliability of these measures. Presumably these measures were collected at baseline and in the online survey in May?
L210. Please confirm how normality of distributions were determined
L212. Are the mean depression score here baseline or during pandemic scores? The indication from the heading is during the pandemic but it is not explicit in text. Clearly depression score did not confirm to normal distribution

L274. The description that depression and depressive symptoms are ‘weakening disorders’ is highly stigmatising and are inappropriate. Many people with depression live highly productive lives and make significant contribution to communities. This must be redressed in your manuscript. Symptom severity varies and while at times may impact functional capabilities, it is not constant.

L282. This statement needs references since some studies show the prevalence of various aspects of psychological distress to be not different to ‘normal’ population severity levels.

L301. Please check the reference cited here. Based on the article abstract there is no mention of depression being examined. If this statement is true then you have used second referencing and the primary reference should be cited. Also 1 reference from a single study is not ‘well documented’

L307. The tests of muscular fitness are markedly different (isometric back extensor and handgrip strength) to those used in the present study. Are there studies with more comprehensive evaluations of muscular fitness (not musculoskeletal, which is not the correct term) for comparison?

L324. The authors seem to interchange the terms exercise and physical activity. The definitions are different and the authors must differentiate between them, especially when considering pulmonary capacity and flexibility.

L348. The reference to a average public effect is overstated since the sample is confined to students at 2 Universities in China

Reviewer 3 ·

Basic reporting

This is an interesting investigating the association between physical fitness during quarantine and depression. Although the clinical importance is well stated, some points need to be addressed and further discussed to make it suitable for publication. The professional use of the English language should be checked by a native English speaker.

Experimental design

1. The statistical methods need to be revised.
a. The authors wrote that ‘The primary outcome of the present study was depression during COVID-19 pandemic’, however, further details need to be provided. Please add if depression would be treated as a categorical or continuous outcome, and the way to define it.
b. The authors stated that Wilcoxon signed-rank tests were used for assessment in mean difference between sexes of categorical variable. However, this is not appropriate. Wilcoxon signed-rank test is a non-parametric test used to compare two correlated samples. To compare categorical variables between gender, chi-square test or fisher’s exact test should be performed.
c. The authors mentioned that ‘The independent relationships of prior physical fitness factors and BDI-II depression score or prevalence of depression during the pandemic were assessed using multivariate linear and binary logistics regressions accordingly.’ Firstly, it is suggested that the authors to add the definition of outcome(eg., is the linear regression based on the score change or absolute number), otherwise it is confusing how the models were performed; Secondly, the authors only added ‘prior physical fitness factors’ as covariates, can the authors confirm if physical fitness factors during the COVID-19 was added? Thirdly, if the authors run two sets of models, it should be described separately to avoid confusion. Lastly, ‘logistics’ should be revised as ‘logistic’.
d. The authors mentioned that ‘a p value <0.20 in the univariate analyses were included in the multivariate linear and binary logistic regression analyses.’ However, no description of univariate analyses were included in statistical section. This should be added.
e. ‘Potential nonlinear effects of decreases versus increases in each variable were evaluated by modeling changes in indicator categories’, here the statement is different from the above ‘prior physical fitness factors were added’. Was the physical fitness and patient characteristics measured in ‘mean change’ or ‘absolute numbers’? Also the ‘modelling changes’ is not specific enough.
f. ‘multivariate’ should be revised as ‘multivariable’, since multivariable is more used for the analysis with one outcome and multiple independent variables, while multivariate is used for the analysis with more than 1 outcomes and multiple independent variables.
g. ‘a two-tailed alpha level of 0.05 was considered significant’, it is suggested to revised it as ‘was considered to be statistically significant’.
2. All the ‘binary logistics regressions’ should be revised as ‘logistic regression’ throughout the paper.
3. It wasn’t clear if the BDI-II was measured once or twice. Was the questionnaire performed only in May 2020? It is suggested for the authors to add a flowchart.

Validity of the findings

1. In the abstract-results section, the authors stated that ‘fitness were independently and inversely associated with depression across sexes’. However, this statement is not accurate, when stating something is different across a group, statistically we normally mean the interaction term is significant. Here it seems like the authors actually mean fitness were associated with depression for the overall population, so it is suggested for the authors to revise this statement throughout the paper.
2. In table 1, for the ‘mean difference’, the direction of calculation should be specified.
3. In table 2, the coefficients obtained from univariate linear regression model should be reported instead of mean. The digits of p-values should be consistent to 3 digits. Same for table 3 and table 4.

---

## Round 0.2 · Minor Revisions

Still pending some minor modifications suggested by one of the reviewers.

Reviewer 1 ·

Basic reporting

Tha authors have properly addressed my requests.

Experimental design

It is fine the way it is.

Validity of the findings

This has been improved.

Additional comments

Tha authors have properly addressed my requests.

Reviewer 2 ·

Basic reporting

The manuscript is generally well written and clear. Even with substantial revisions, it remains well structured with no fatal flaws. My previous concerns regarding references have been addressed

Experimental design

The design is generally consistent with the aims. This original primary research fits within the scope of the journal. The previously understated significance of this study has now been addressed. I am pleased the other reviewers have commented on the statistical analysis and these appear to have been adequately addressed.

Validity of the findings

This is markedly improved in this revision and the conclusions have been tempered to be more conservative.

Additional comments

I have no further comments

Reviewer 3 ·

Basic reporting

no comment

Experimental design

no comment

Validity of the findings

1. L226: the primary outcome should be the 'score change', but not the score itself. 'Prevalence of depression' should be revised as 'presence of depression'.
2. For table3 and table4, the coefficient and its 95% CI from univariate/multivariable linear regression should be reported instead of the mean change. For table2, please add 95%CI for each covariate.

Additional comments

Thanks the authors for addressing my questions.

---

## Round 0.3 · accepted · Accept

All the reviewers' concerns have been correctly addressed.

Reviewer 3 ·

Basic reporting

no comment

Experimental design

no comment

Validity of the findings

no comment

Additional comments

The authors have adequately addressed my questions. Thank you.